# Stemming Tumoral Growth: A Matter of Grotesque Organogenesis

**DOI:** 10.3390/cells12060872

**Published:** 2023-03-11

**Authors:** Marisa M. Merino, Jose A. Garcia-Sanz

**Affiliations:** 1Department of Biochemistry, Faculty of Sciences, University of Geneva, 1205 Geneva, Switzerland; 2Department of Molecular Biomedicine, Centro de Investigaciones Biologicas Margarita Salas, Spanish National Research Council (CIB-CSIC), 28040 Madrid, Spain

**Keywords:** organogenesis, tumorigenesis, stem cells, cancer stem cells, cell competition, scaling, size, growth

## Abstract

The earliest metazoans probably evolved from single-celled organisms which found the colonial system to be a beneficial organization. Over the course of their evolution, these primary colonial organisms increased in size, and division of labour among the cells became a remarkable feature, leading to a higher level of organization: the biological organs. Primitive metazoans were the first organisms in evolution to show organ-type structures, which set the grounds for complex organs to evolve. Throughout evolution, and concomitant with organogenesis, is the appearance of tissue-specific stem cells. Tissue-specific stem cells gave rise to multicellular living systems with distinct organs which perform specific physiological functions. This setting is a constructive role of evolution; however, rebel cells can take over the molecular mechanisms for other purposes: nowadays we know that cancer stem cells, which generate aberrant organ-like structures, are at the top of a hierarchy. Furthermore, cancer stem cells are the root of metastasis, therapy resistance, and relapse. At present, most therapeutic drugs are unable to target cancer stem cells and therefore, treatment becomes a challenging issue. We expect that future research will uncover the mechanistic “forces” driving organ growth, paving the way to the implementation of new strategies to impair human tumorigenesis.

## 1. Early Metazoans and Division of Labour

The emergence of metazoan life can be traced back to 700–800 million years ago [1]. Primitive metazoans probably evolved from single-celled organisms which found the colonial system to be a beneficial organization [2,3,4,5,6]. Over the course of evolution, these primary colonial organisms increased in size, and division of labour among the cells became a remarkable feature [7]. Cells within the colony specialized for different functions in order to obtain the required resources, perhaps while reducing conflicts among the members of the colony [8]. This evolutionary process led to a higher level of biological organization, which in current complex organisms has developed into anatomical units, with specialized cells that form the biological organs, which perform specific functions [2,3,4,5,6].

Porifera (commonly referred to as sponges) are thought to be the most ancient, extant metazoans on earth, and therefore provide insights into the earliest processes in metazoan evolution [2,9]. These living fossils already carry some signalling pathways, including Wnt, TGF-Beta, and Hedgehog [10], which are conserved across metazoan life. The sponge’s anatomy is formed by two distinct epithelial cell layers, with a few cell types showing different degrees of motility [11]. Indeed, sponges were the first organisms in evolution to show organ-type structures. Sponge epithelia are polarised, sealing and controlling the passage of solutes and therefore their internal milieu. This proto-skin might be evolutionary the origin of the first biological organ in the known life, allowing biochemical communication among cells and probably setting the grounds for complex organs to evolve [12].

## 2. The Emergence of Tissue-Specific Stem Cells

Throughout evolution, and concomitant with organogenesis, is the appearance of tissue-specific stem cells in the animal and plant kingdoms [13]. Stem cells sustain cell renewal, replace and regenerate damaged tissues, maintaining organ function and homeostasis [14,15,16]. Historically, the term “stem cell” was coined by the German physician and zoologist Ernest Haeckel (1834–1919). While recording his findings, he extraordinarily blended science and art by hand-drawing the organisms he observed, from mountaintops to oceans [17]. Curiously, Haeckel introduced the term with an evolutionary connotation. As a supporter of Darwin’s theory of evolution, he described phylogenetic trees, which he called “stem trees”, as representing the evolution of organisms from common ancestors. In this context, he used the term “stem cell” to refer to the unicellular ancestor of all multicellular organisms, as well as the fertilized egg, which gives rise to all cell types in living organisms [18].

Despite this early usage of the term, it was not until the 1960s that the Canadian scientists Ernest A. McCulloch and James E. Till characterized what we now refer as a stem cells, and developed an assay to analyse the potential of early blood-forming progenitor cells from bone marrow to self-renew and differentiate into several blood cell types (Figure 1). These pioneering studies described the hallmark properties of stem cells, characterized by the ability to self-renew and differentiate in a hierarchical organization [19,20,21]. Going back to the primitive metazoan organisms, sponge stem cells already carry out roles in cellular specialization and renewal, which might explain the powerful regeneration capacity of these living systems [22,23]. In the case of higher organisms, stem cell potency is reduced over time during specialization [15]. Organ formation in the embryos of these organisms requires the presence of tissue-specific stem cells, which later give rise to the organ-specific cell hierarchy driven by growth and patterning signals [15,24]. In the adult stages, organ architecture and functionality are maintained by somatic stem cells, which are found in low frequencies in most adult tissues [25,26,27,28,29,30,31].

Thus, evolution gave rise to multicellular living systems formed by distinct anatomical structures (i.e., organs) which perform one or more physiological functions. This anatomical organization is the result of a cellular hierarchy established during development, which maintains required features through adulthood. As a result of this hierarchy, organ systems communicate and work together to generate successful living individuals. This setting seems to be a constructive role of evolution; however, what happens when rebel cells against the current regulation came up into this setting?

## 3. Tumoral Growth: From Proto-Oncogenes and Hallmarks of Cancer, to the Cancer Stem Cell Hypothesis

In certain diseases, whether they occur during development or adulthood, cells within tissues can take over the existing molecular mechanisms for other purposes. One of these examples is tumorigenesis. Initial studies in the cancer field showed the role of proto-oncogenes and oncogenes in tumoral growth [32,33]. Later, in 2OOO, the hallmarks of cancer were discussed by Hanahan and Weinberg, suggesting that most cancers acquire a similar set of functional capabilities, although by using different molecular mechanisms [34]. These capabilities include, amongst others, *(i)* limitless replicative potential; *(ii)* self-sufficiency in growth signals; *(iii)* tissue invasion and metastasis; and *(iv)* evasion of apoptosis [34,35]. In an attempt to reconcile all of these facts, a new hypothesis explaining the bases of tumoral growth was subsequently formulated [36]. The cancer stem cell hypothesis suggested that within the tumour, there is a small subpopulation of cells with stem cell properties, which drive tumoral growth [36]. Cancer stem cells (CSCs) were initially described in leukaemia, and afterwards in solid tumoral tissues, including mammary gland and brain tumours [37,38,39]. Despite the initial controversies on the existence of CSCs, the presence of CSCs was later demonstrated in in vivo studies [40], which identified CSCs in adenomas, melanomas, gliomas and mammary gland tumours [41,42,43,44] (Figure 1).

## 4. Tumours as Caricatures of Dysfunctional Organs

Nowadays, it is established that CSCs are at the top of a hierarchy. These cells divide either symmetrically, amplifying the CSC population, or asymmetrically, generating transient amplifying cells with high proliferative potential and giving rise to more differentiated cell populations and tumoral heterogeneity. In addition, CSCs are the root of metastasis generation, therapy resistance, and relapse [45,46,47,48]. Indeed, some aspects of tumour development greatly resemble the processes of organ development. CSCs can control tumoral growth and also have differentiation capacity, generating aberrant organ-like structures which show reminiscent patterning from the original organs. However, this tumoral patterning is mostly observed in early cancer stages. Over time and contrary to *wild-type* organ development, cells within the tumour acquire fewer differentiated traits, somehow showing a process of organ devolution [48,49]. It is worth mentioning that tumours within a living individual perform different functions and communicate with other organs [49], though most of these functions are likely not well aligned with the other organ systems in terms of organismal survival [49]. Tumour-to-organ communication triggers several systemic responses, including: suppression of the immune system, coagulation abnormalities, and changes in metabolism. One striking effect of tumour-to-organ communication is cachexia, a wasting disorder which causes severe weight loss and fatigue [50,51]. Cachexia directly causes up to 30% of cancer deaths and is induced by factors secreted by the growing tumours [49]. Therefore, these systemic tumoral effects are the cause of a high percentage of cancer deaths, rather than the primary tumoral growth or metastasis itself [49].

## 5. Cell Competition in Primary and Secondary Cancer Growth

Several studies confirm that tumours can develop from single clones which acquire a set of mutations driving overgrowth [52,53]. These mutations are of diverse origins: they can be inherited in the autosomal or mitochondrial DNA, induced by chemical carcinogens or environmental factors, or modulated by the individual’s microbiome [54,55,56,57,58,59,60,61] (Figure 1). Tumours, while growing, increase by volume and thus the contact area with the *wild-type* host tissues also increases. At this tumour–*wild-type* interface, tumoral cells battle against host cells in order to make “territorial gains”. During this battle, host cells are usually eliminated from the tissue undergoing apoptosis, and the speed at which the host cells are eliminated correlates with the tumoral growth rate [41,62,63,64,65,66,67,68]. This phenomenology of cell competition was initially found in *Drosophila* growing tissue when analysing ribosomal mutations (i.e., *Minute*) using genetic mosaics [49]. The so-called unfit cells were eliminated from the growing organ in the presence of the so-called fit cells [69,70]. Currently, we know that cell competition principles are conserved from *Drosophila* to humans, playing multiple physiological roles during development, adulthood, disease, and ageing [41,62,63,64,65,71,72,73,74,75,76,77,78,79,80,81,82,83,84,85,86].

In the literature, a great number of studies show that the bases of these competitive behaviours are of different natures [14,41,62,63,69,71,72,73,77,81,87,88,89]. Cells can battle for growth factors and also space; furthermore, signalling levels and growth rates seem to play a big role in the game [62,65,69,77,81,87,88,90,91,92,93,94,95,96,97,98,99]. For instance, during organ growth in *Drosophila,* and in vertebrates, cells with “wrong” morphogen signalling levels are eliminated from the growing field in order to maintain *wild-type* size or pattern [87,88,99].

Primary tumoral growth happens within the original tissue/organ where it was initiated. However, these primary tumoral cells can acquire migratory capabilities in order to conquer overseas territory. During the metastatic cascade, cancer cells leave the primary tumour, travel through the bloodstream or lymphatic system, and reach distant organs, then develop secondary tumoral outgrowth or metastasis [100,101]. Remarkably, tumoral cells do not metastasize in a random manner. Cancer metastases are selectively distributed depending on the cancer type and subtype. Some examples of organ-specific metastasis include the preference of pancreatic and uveal cancers to form metastasis to the liver, while prostate cancer often metastasizes to bone [100,102,103,104]. It is shown that some tumours over-express chemokine receptors, which are functional and therefore trigger the migration of tumour cells towards the sites where the ligands for the particular receptor expressed in the tumour cells are synthesized [105]. For instance, tumours expressing the chemokine receptor CCR9 tend to generate metastasis on the small intestine [105,106,107,108,109,110,111,112].

However, the mechanisms that cancer cells exploit in order to cause secondary overgrowths in targeted organs are largely unknown, and it is still a major open question in the field [49,100]. A plausible hypothesis is that during the metastatic process, migrating cancer cells, which carry reminiscent signalling profiles from their original organs, must find a field that is compatible with their own signalling profile in order to grow. Otherwise, cancer cells within an incompatible field will be outcompeted and eliminated, a process which we might call “metastatic competition” (Figure 2A,B). Perhaps, this kind of competition could also engage spatial constraints from the target field. Actually, this hypothesis is consistent with the “seed and soil theory” proposed by Stephen Paget in 1889. Paget compared the migrating cancer cells to seeds which are randomly distributed, but will grow solely in congenial soil [100,113,114].

## 6. Scaling Phenomena during Development, Adulthood, and Tumorigenesis

It is striking how developing systems remain proportional while growing. In the case of human development, the size of our hands scales with the size of our growing body. These proportionalities were already discussed long ago; however, the scaling mechanisms were not known [115]. Indeed, during organ growth, these proportionalities can be controlled by morphogen gradients. *Drosophila* research has largely contributed to our understanding of the molecular mechanisms regulating scaling in organ growth. For instance, in the developing *Drosophila* wing, the Decapentaplegic (Dpp) morphogen regulates growth and patterning phenomena [116,117,118,119,120,121,122,123]. Dpp spreads from a source, located at the centre of the tissue, and forms a concentration gradient into the target cells, which can be measured by its particular decay length (λ) [116,120,124]. It has been shown that during wing growth, the λ of the Dpp gradient correlates with the size of the tissue. That is, the Dpp gradient scales with tissue size [116,120]. To fulfil this job, there is a mechanism which tunes the λ during organ growth, keeping the gradient and tissue proportional [87,88,116,125,126,127]. In this particular organ (i.e., wing), Dpp signalling scaling failure leads to defects in tissue size and pattern formation [87,88,116,126,128,129].

It is noteworthy that organ scaling happens not only in developing systems, but also during adulthood. A peculiar scenario is the case of regenerative growth, by which organisms replace or restore tissues and organs. This mechanism ensures that the final size of the regenerating tissue or organ remains proportional with morphological patterns of the adult individual [130,131]. However, throughout evolution, there is loss of regenerative growth [132]. A second example of this is organ scaling in human adulthood. Several reports show that organ weight can correlate with total body weight, additionally showing different means of organ weight between females and males [24,133,134], most probably due to modulation of the total number of cells per organ, although, the mechanistic bases of these proportionalities remain unknown. Analogously, the size differences observed between female and male organs can also be found in tumoral growth. Interestingly, studies describe that tumour sizes can differ significantly between females and males, exhibiting bigger tumour sizes in males than females [135,136,137]. This fact does not appear to be trivial, and at the moment we might not have a biological explanation for this phenomenon.

## 7. Concluding Remarks and Future Views

A widespread feature among different cancer types is the association of tumour size with patient outcomes [138,139,140]. Generally, bigger tumour sizes correlate with worse outcomes [138,139,140]. At present, most clinically used therapeutic drugs target cancer cells with high proliferation rates; however, CSCs show SC features, including slow proliferation rates [45,48] and usually they remain unaffected by these drugs. Therefore, treatment-wise, this becomes a challenging issue: pruning the branches from the mistletoe does not kill the parasite, perhaps it will make it grow more vigorously instead.

The literature describes the same order of magnitude in the number of cells per volume within *wild-type* epithelial tissues and tumours of epithelial origin [141,142,143,144] (~10^8^ cells/cm^3^). An open possibility is to tackle the mechanisms controlling tumour size. During tumoral growth, different clones of cells are generated, which might reflect reminiscent patterning cues. Within the tumoral field, as happens during *wild-type* organ growth, clones of cells can act as sources of morphogens and other secreted factors, cooperating with each other and thus controlling tumour size [122,126,128,145] (Figure 3A). Morphogens and other secreted factors can be produced by tumoral cells, secreted, and spread throughout the tissue [122,126,128,145], then acting in a paracrine fashion on other cells by regulating growth and pattern formation [122,126,128,145]. In this context, altering the biophysical properties of the tumoral field could be an option to consider for therapeutic purposes. Hence, by decreasing the range of these secreted factors within the tumoral field, proliferative rates of cancer cells might decrease, and therefore modulate tumour growth (Figure 3A,B). Regarding these issues, we expect that future research will uncover the mechanistic “forces” driving organ growth, which may pave the way to the implementation of new strategies to impair human tumorigenesis.

## Figures and Tables

**Figure 1 cells-12-00872-f001:**
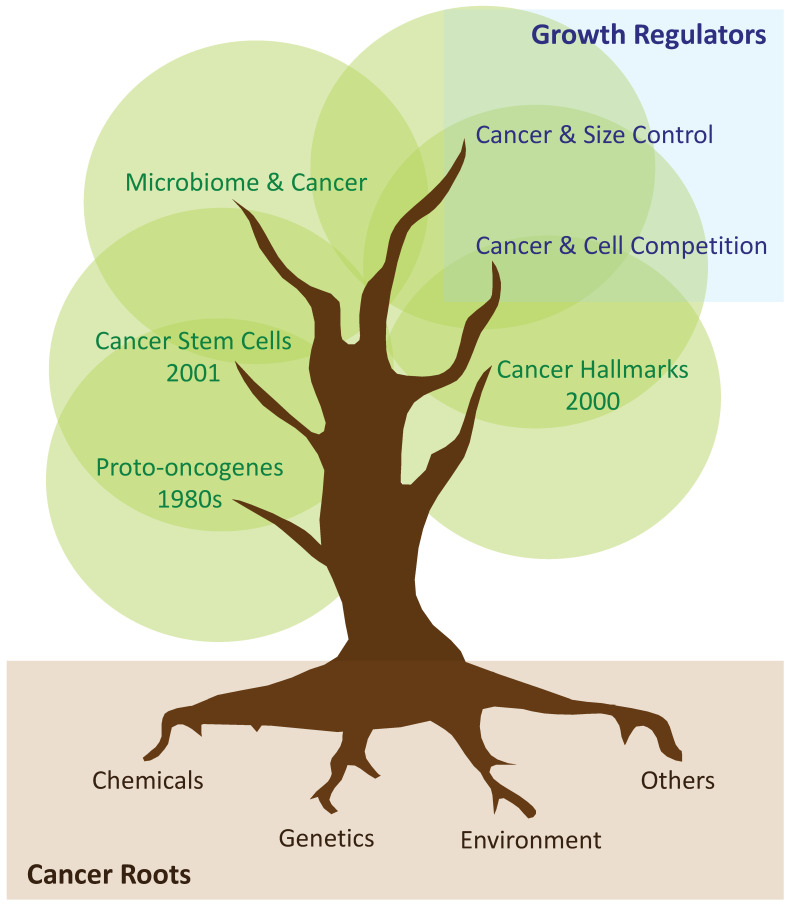
**Cancer roots and growth regulators**. Tree chart depicting known origins of cancer (brown), timeline of key discoveries in cancer research (green), and growth regulators during tumoral growth (blue).

**Figure 2 cells-12-00872-f002:**
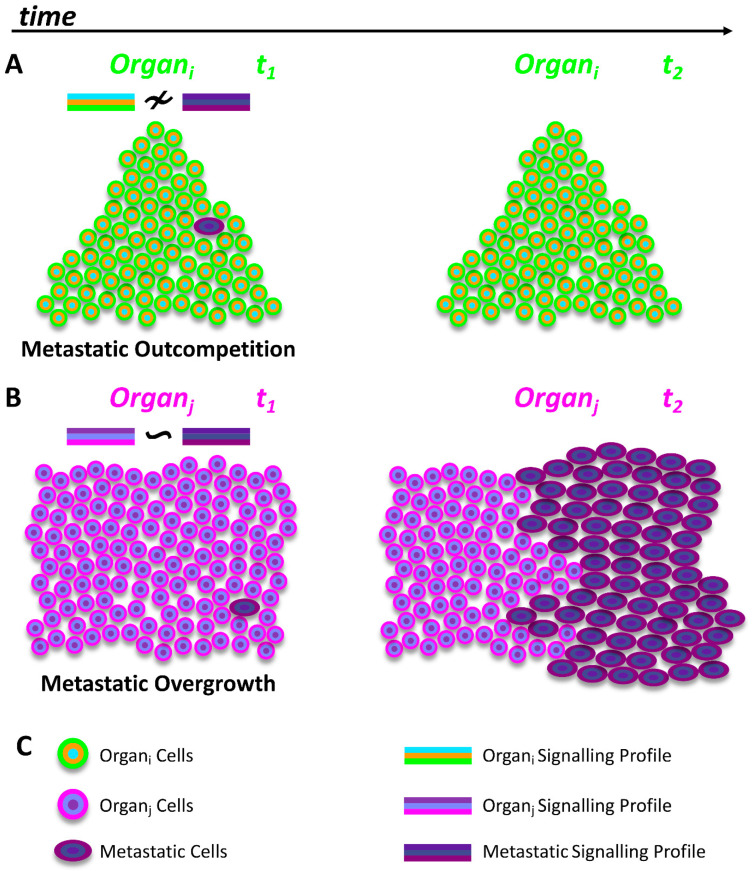
**Metastatic Competition hypothesis and tumoral growth.** (**A**). Metastatic cells with different signalling profiles (≁) than the signalling profiles from the target organs (organ_i_) might be outcompeted and eliminated from the tumoral field, thus abolishing metastatic growths. (**B**). Metastatic cells with similar signalling profiles (~) to the signalling profiles from the target organs (organ_j_) might not be eliminated from the organ microenvironment, allowing them to proliferate and form metastatic overgrowths. (**C**). Signalling profiles and cells from organ_i_ and organ_j_ are depicted in different colours; *t* refers to time.

**Figure 3 cells-12-00872-f003:**
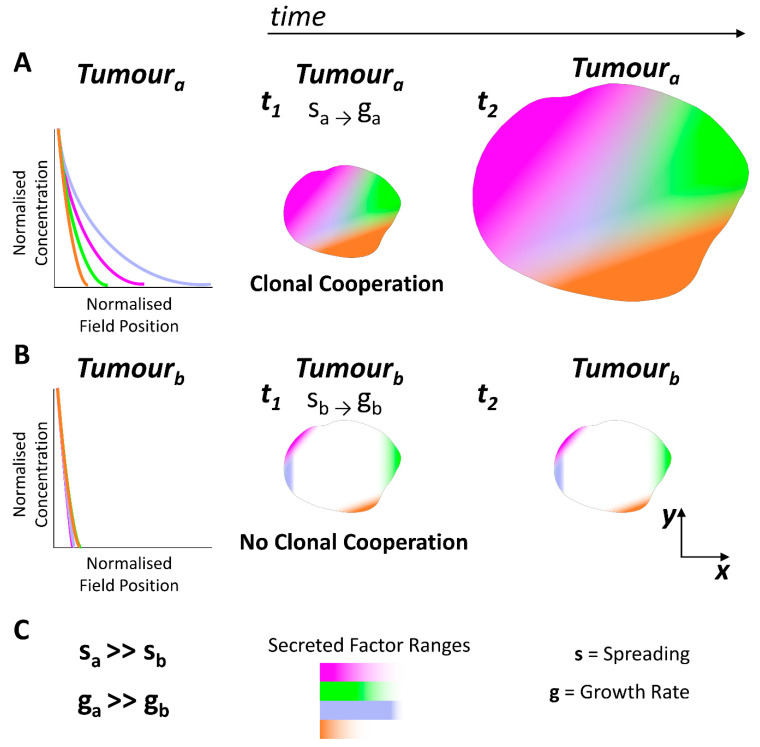
**Clonal cooperation and tumoral growth.** During tumoral growth, different clones/subclones of cancer cells are generated, which can act as secreted factor sources, boosting tumoral growth. The spread of these secreted factors depend on the biophysical properties of the tumoral field. (**A**). Depicted plot showing normalized secreted factor concentration vs. normalized field position within the tumoral field **a**. Different colours depict different secreted factors. Tumoral field **a** (tumour_a_), shows high spreading biophysical properties, increasing the range of these secreted factors throughout the tumoral field. These can lead to clonal/subclonal cooperation, increasing the proliferation of cancer cells and therefore boosting tumoral growth rates (**g**). (**B**). Depicted plot showing normalized secreted factor concentration vs. normalized field position within the tumoral field **b**. Different colours depict different secreted factors. Tumoral field **b** (tumour_b_) shows low spreading biophysical properties; thus, secreted factors spread over small ranges within the tumoral field. Small ranges of secreted factors in the tumoral field can abolish clonal cooperation and decrease tumoral growth rates (g). (**C**). **s**: refers to the spreading ranges of secreted factors; **g**: refers to the tumoral growth rate; *t*: refers to time.

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
