# Peer review of "Stemming Tumoral Growth: A Matter of Grotesque Organogenesis"

_cells, 2023, doi:10.3390/cells12060872_

Round 1

Reviewer 1 Report

Merino and Garcia-Sanz present an interesting views and perspectives on the link between tumoral growth and organongenesis. 

Some suggestions that will enhance the impact of this paper.

1. In the initial texts, a section on tumor hallmarks with expanding views will bring more attention to readers.

2. A analysis of existing data on the number of cancer cells per unit weight of tumor tissues and healthy tissues will open up discussion on tumor organogenesis.

3. A perspectives of tumor organogenesis and microbiome could be explored.

4. Views on the involvement of mitochondrial inheritance and  tumoral organogenesis may be discussed.

5. Discussion on the impacts of chemical carcinogens induced tumors and their growth could be discussed. 

Author Response

Reviewer 1

Merino and Garcia-Sanz present an interesting views and perspectives on the link between tumoral growth and organongenesis. 

We thank the Reviewer for the positive comments on our opinion article.

Some suggestions that will enhance the impact of this paper.

  1. In the initial texts, a section on tumor hallmarks with expanding views will bring more attention to readers.

We have considered to include the section on tumour hallmarks with expanding views in the initial text. If it is fine for the Reviewer we think it is more logical to keep this part in the section: Tumoral Growth: from proto-oncogenes and hallmarks of Cancer, to the Cancer Stem Cell hypothesis (Line 83).

  1. A analysis of existing data on the number of cancer cells per unit weight of tumor tissues and healthy tissues will open up discussion on tumor organogenesis.

In the Revised version of the manuscript we have included the data available in the literature which refer to the number of cells per tissue volume. Indeed, these reports do not show big differences in the number of cells per tissue volume comparing epithelial wildtype tissues and epithelial tumours1-4. It should be noted that for many tumours there is no data available on the number and size of tumoral cells per weight unit of tumoral tissue.

We now write:

Line 201. “most probably by modulating the total number of cells per organ”

Line 216. The literature describes the same order of magnitude in the number of cells per volume within wildtype epithelial tissues and tumours of epithelial origin1-4 (~108 cells/cm3)”.

  1. A perspectives of tumor organogenesis and microbiome could be explored.

We are well aware of the role of microbiome on carcinogenesis, however, the links of microbiome with CSC, cell-cell competition and organ size have, as far as we are aware, still to be made. Considering the Reviewer's suggestion, we have now included in the text the link between microbiome and cancer (see below).

  1. Views on the involvement of mitochondrial inheritance and tumoral organogenesis may be discussed.

We are aware that the mitochondria of an organism are maternally inherited, and well aware that there is a role of the mitochondrial genome on tumorigenicity, however, the link between mitochondrial inheritance and organ size control are not well established. Considering the Reviewer's suggestion, we have now included in the text the link between mitochondrial DNA and cancer (see below). Furthermore, in the Revised version we have included a new Figure (New Figure 1). 

  1. Discussion on the impacts of chemical carcinogens induced tumors and their growth could be discussed. 

Considering the Reviewer's suggestions 3, 4 and 5, we have now included a text describing the origins of the primary cancer overgrowths. These origins include, mutations in the autosomal or mitochondrial DNA, chemical carcinogenesis, environmental factors and individual's microbiome5-12. Also, in the Revised version we have included a new Figure (New Figure 1).  

In the Revised version we write:

Line 127. “These mutations are of diverse origin, they can be inherited in the autosomal or mitochondrial DNA, induced by chemical carcinogens or environmental factors and modulated by the individual's microbiome5-12”.

REFERENCES

  1. Wunderlich, S.; Haase, A.;  Merkert, S.;  Jahn, K.;  Deest, M.;  Frieling, H.;  Glage, S.;  Korte, W.;  Martens, A.;  Kirschning, A.;  Zeug, A.;  Ponimaskin, E.;  Gohring, G.;  Ackermann, M.;  Lachmann, N.;  Moritz, T.;  Zweigerdt, R.; Martin, U., Targeted biallelic integration of an inducible Caspase 9 suicide gene in iPSCs for safer therapies. Mol Ther Methods Clin Dev 2022, 26, 84-94.
  2. Tian, Y.; Zhao, S.;  Zheng, J.;  Li, Z.;  Hou, C.;  Qi, X.;  Kong, D.;  Zhang, J.; Huang, X., A stereological study of 3D printed tissues engineered from rat vaginas. Ann Transl Med 2020, 8 (22), 1490.
  3. Holt, P. G.; Degebrodt, A.;  Venaille, T.;  O'Leary, C.;  Krska, K.;  Flexman, J.;  Farrell, H.;  Shellam, G.;  Young, P.;  Penhale, J.; et al., Preparation of interstitial lung cells by enzymatic digestion of tissue slices: preliminary characterization by morphology and performance in functional assays. Immunology 1985, 54 (1), 139-47.
  4. Del Monte, U., Does the cell number 10(9) still really fit one gram of tumor tissue? Cell Cycle 2009, 8 (3), 505-6.
  5. Waarts, M. R.; Stonestrom, A. J.;  Park, Y. C.; Levine, R. L., Targeting mutations in cancer. J Clin Invest 2022, 132 (8).
  6. Trosko, J. E., The Concept of "Cancer Stem Cells" in the Context of Classic Carcinogenesis Hypotheses and Experimental Findings. Life (Basel) 2021, 11 (12).
  7. Sepich-Poore, G. D.; Zitvogel, L.;  Straussman, R.;  Hasty, J.;  Wargo, J. A.; Knight, R., The microbiome and human cancer. Science 2021, 371 (6536).
  8. Park, E. M.; Chelvanambi, M.;  Bhutiani, N.;  Kroemer, G.;  Zitvogel, L.; Wargo, J. A., Targeting the gut and tumor microbiota in cancer. Nat Med 2022, 28 (4), 690-703.
  9. Kim, M.; Mahmood, M.;  Reznik, E.; Gammage, P. A., Mitochondrial DNA is a major source of driver mutations in cancer. Trends Cancer 2022, 8 (12), 1046-1059.
  10. Dong, J.; Wong, L. J.; Mims, M. P., Mitochondrial inheritance and cancer. Transl Res 2018, 202, 24-34.
  11. Dietrich, C.; Weiss, C.;  Bockamp, E.;  Brisken, C.;  Roskams, T.;  Morris, R.;  Oesch-Bartlomowicz, B.; Oesch, F., Stem cells in chemical carcinogenesis. Arch Toxicol 2010, 84 (3), 245-51.
  12. Lewandowska, A. M.; Rudzki, M.;  Rudzki, S.;  Lewandowski, T.; Laskowska, B., Environmental risk factors for cancer - review paper. Ann Agric Environ Med 2019, 26 (1), 1-7.

Reviewer 2 Report

Dear authors,

The opinion on " Stemming Tumoral Growth: A Matter of Grotesque Organo- 2 genesis" was written well with adequate supportive references.  It was easy to follow the write-up and understanding.  

Upon careful reading, there was no major comments to correct the article.  However if you can contribute a flow chart  having molecular evolutionary  evidence  for CSCs'  as a supportive document may attract more readers. 

Author Response

Reviewer 2

Dear authors,

The opinion on " Stemming Tumoral Growth: A Matter of Grotesque Organo- 2 genesis" was written well with adequate supportive references.  It was easy to follow the write-up and understanding.  

We thank the Reviewer for the positive comments on our opinion article.

Upon careful reading, there was no major comments to correct the article.  However if you can contribute a flow chart having molecular evolutionary evidence for CSCs' as a supportive document may attract more readers.

Following the Reviewer's suggestion in the Revised version of the manuscript we have now included a New Figure (New Figure 1) showing a flow chart of the molecular evolutionary evidence for CSCs' and also other relevant phenomena modulating tumoral growth.